# Evaluation of the Pullout Behavior of Pre-Bored Piles Embedded in Rock

**DOI:** 10.3390/ma14195593

**Published:** 2021-09-26

**Authors:** Kyungho Park, Daehyeon Kim, Gyudeok Kim, Wooyoul Lee

**Affiliations:** 1Department of Civil Construction Engineering, Chosun College of Science & Technology, Gwangju 61453, Korea; parkgeo@cst.ac.kr; 2Department of Civil Engineering, Chosun University, Gwangju 61452, Korea; 3Department of Construction, KL Engineering Co., Jangsung 57241, Korea; kgd125@hanmail.net (G.K.); ewoo10@naver.com (W.L.)

**Keywords:** dry process caisson tube method cofferdam, skin friction, pre-bored pile, pull-out load test, grout

## Abstract

The subject of this study is dry process caisson tube method cofferdam (hereinafter called C.T cofferdam). This C.T cofferdam is designed to use the skin friction of the drilled shaft embedded into the rock for stability of buoyancy. A pre-bored pile embedded in the bedrock was pulled out due to the buoyancy of the C.T cofferdam at the pier (hereinafter called P) 2 of the OO bridges under construction, to which this was applied. In this study, in order to solve this problem, the adhesion force applied with the concept of skin friction and the pre-bored pile of drilled shaft according to domestic and foreign design standards were identified; the on-site pull-out load test was used to calculate the pull-out force; and the skin friction of the drilled shaft and pre-bored pile embedded into the bedrock were compared and analyzed. In addition, the pull-out behavior of the pre-bored pile embedded in the bedrock was analyzed through numerical analysis. The adhesion strength tested in the lab was 881 kN for air curing of concrete and 542 kN for water curing of concrete, and the on-site pull-out test result was 399.7 kN. As a result of the numerical analysis, the material properties of the grout considering the site conditions used revealed that the displacement of the entire structure exceeded the allowable limit and was unstable. This appears to have lowered the adhesion strength due to construction issues such as ground complexity and both seawater and slime treatment, which were not expected at the time of design.

## 1. Introduction

### 1.1. Research Background and Purpose

End-bearing and skin friction behavior of piles subjected to axial loads are important factors in predicting the bearing capacity of structures. However, the ISO 19902 [1] regulations on offshore steel structures do not suggest an accurate calculation equation for the evaluation of friction-bearing capacities at the tip and circumferential surface of piles embedded in bedrock.

The Korean structural foundation design standard (Ministry of Land, Infrastructure, and Transport [2]) briefly mentions the end-bearing capacity for compression but does not suggest any formula for calculating the skin friction of the pile foundation embedded in the rock. Therefore, when designing offshore structures, the equation for calculating the skin friction of drilled shafts embedded in the bedrock, which is the most similar mechanism to the required calculation, is applied.

Due to the fact that the bearing capacities of a drilled shaft and pre-bored pile are very different, it is easy to overestimate or underestimate the skin friction of the pile. Therefore, due to incorrect evaluation, the safety of the structure may be impaired and errors may occur in the prediction of the behavior of the pile foundation. In order to construct the structure safely and economically, it is necessary to study the support behavior of the pre-bored pile embedded in the bedrock.

C.T cofferdam is designed to use the skin friction of drilled shafts embedded in the rock to ensure buoyancy stability. However, due to the buoyancy of C.T cofferdam, a problem occurred in which the pre-bored pile was pulled out. To solve this problem, firstly, we analyzed the skin friction of the drilled shaft according to domestic and foreign design standards. Secondly, a lab test was performed on the adhesion force, in which the concept of a pre-bored pile was applied. Thirdly, the pull-out force was calculated through the pull-out load test (ASTM D3689-07 [3]) and both the skin friction of the drilled shaft and pre-bored pile embedded in the bedrock were compared and analyzed. Finally, the pull-out behavior of the pre-bored pile embedded in the bedrock was analyzed through numerical analysis.

### 1.2. Research Trends

Many researchers have been conducting studies on the bearing capacity of piles for a long time. In the case of soft ground and weathered rock, the basic design theory is established based on the results of field tests. However, past studies on pile foundations embedded in bedrock mainly focused on drilled shafts and there are very few studies on pre-bored piles.

When calculating the ultimate bearing capacity of a drilled shaft embedded into rock, design standards and bearing capacity calculation equations that were proposed overseas (FHWA [4]; FHWA [5]; and Canadian Geotechnical Society [6]) are applied.

In Korea, overseas design standards and bearing capacity calculation equations were modified to match domestic standards and were applied to the structural basic design standards and road bridge design standards (Ministry of Land, Infrastructure, and Transport [2]).

Previous overseas studies on skin friction of pre-bored piles embedded in bedrock were reviewed. Maertens [7] evaluated the performance of piles by installing steel pipe piles on weathered rocks but they only evaluated limited field conditions.

Shakir-Khalil [8], Shakir-Khalil [9], and Nezamian et al. [10] studied the adhesion strength and adhesion behavior between concrete and steel pipes when steel pipes are filled with concrete.

Moon et al. [11], a previous study in Korea, determined the skin friction of steel pipe piles embedded in bedrock by the skin friction between the surface of the steel pipe and the grout. Moon et al. [12] confirmed that the adhesion strength of the skin friction of the pre-bored pile embedded in the bedrock was about 1 MPa through a model test. Moon and Park [13] also evaluated the effect of the grout mixing ratio on the skin friction of steel pipe piles embedded in bedrock. Regardless of W/C, in the case of grout, the adhesion strength was about 540~560 kPa.

Kim et al. [14] proposed that the load-settlement curves and axial load distributions of piles, as well as the load-transfer curves (t-z curves), are obtained on the main surface of a steel pipe pile. In addition, the N-value obtained from the standard penetration test for the strata, the compressive strength of the bedrock, and the soil-cement formed by the injected grout were confirmed to be affected by uniaxial compressive strength.

Kim and Kim [15] evaluated the skin friction characteristics of the pre-bored pile embedded in bedrock through the dynamic load test of the pile foundation. Chae et al. [16] proposed an equation for calculating the axial bearing of SDA (separated doughnut auger) pre-bored piles embedded in bedrock.

According to the previous domestic and foreign studies related to the calculation of the pull-out force of pre-bored piles embedded in bedrock, the pull-out force of the pre-bored pile is mostly related to the adhesion force of the grout. However, the related design standards are not clear, thus, in practice, the pull-out standards of drilled shafts are applied. In addition, most of the previous studies were limited to the skin friction of steel pipe piles embedded in bedrock among pre-bored piles.

Therefore, in this study, theories and empirical equations similar to the behavior of pre-bored piles were reviewed among the design standards for drilled shafts penetrated into bedrock. What sets this study apart from previous studies is the analysis of the pull-out force behavior of composite piles (upper: steel pipe, lower: reinforced concrete), in which the lower reinforced concrete piles are embedded in bedrock among the pre-bored piles.

## 2. Design-Bearing Capacity of Piles

### 2.1. Estimation of The Bearing Capacity of Piles

As shown in Equation (1), the bearing capacity of the pile consists of the end-bearing capacity and the skin friction force, and the resistance force of the two elements occurs completely at the same time.
(1)$$Q_{u}=Q_{s}+Q_{p}=f_{s}A_{s}+q_{p}A_{p}$$
where $f_{s}$ is the unit skin friction force (kN/m^2^); $A_{s}$ is the major surface area of pile (m^2^); $q_{p}$ is the unit end-bearing capacity (kN/m^2^); and $A_{p}$ is the pile cross-sectional area (m^2^).

Theoretical Equation (1) for obtaining bearing capacity was developed by many scholars for safer and more economical construction after Rankine published the earth pressure theory. Although the understanding of the overall pile has been improved through these efforts, there are many variables affecting the bearing capacity, thus there is a limit to deriving a reliable bearing capacity calculation equation.

In classifying the bearing capacity calculation equations proposed so far, there is the static-bearing capacity calculation method and the method by static loading and dynamic loading tests. The static-bearing capacity equation, based on experience or theory, is mainly used during the design process, although its reliability is somewhat low. In the construction process, the most reliable on-site in situ test methods, such as the static load test and dynamic load test, are mainly applied, although it takes a great deal of time and are costly.

### 2.2. Skin Friction of Pile

Figure 1 shows the skin friction acting on the skin of the pile embedded in the sandy soil. The ultimate unit skin friction force is calculated using Equation (2) based on the theory considering the friction generated on the surface of a rigid body.
(2)$$f_{s}=Kp_{0}tan\delta$$
where K is the horizontal earth pressure coefficient; $p_{0}$ is the effective overburden payload (kN/m^2^); and tanδ is the internal friction angle (°).

Assuming that the sand between the ground and pile is in the ultimate fracture state, the friction angle was estimated to be the same as the residual friction angle $\emptyset_{\tau cs}$ of the sand rather than the unit weight of the ground and the material of the pile. This value is not significantly different from the value according to the method of Vesic [17].

Meyerhof [18] determined the coefficient of earth pressure. The horizontal earth pressure coefficient considers only the horizontal displacement of the pile foundation. This horizontal displacement affects the surrounding ground so that the soil is compacted and the maximum displacement occurs at the point of contact between the pile and the soil.

Meyerhof [19], Poulos and Davis [20], Vesic [21], and Berezantzev [22] proposed a theoretical hydrostatic-bearing capacity calculation method that divided end-bearing capacity and skin friction force. In order to calculate bearing capacity according to the theoretical equation, various ground characteristics are required. In addition, both lab and field tests are required to calculate this. Values that cannot be obtained from lab and field tests should be assumed and applied. Therefore, when using the theoretical equation, uncertainty exists because it assumes variables that cannot be obtained from lab and field tests. It is economically disadvantageous because various soil tests are required. There is also the problem of poor accuracy when applying it to the design. Table 1 shows the theoretical equations of the hydrostatic-bearing capacity currently applied in Korea and abroad.

The skin friction of clay soil was calculated by the α coefficient method and β coefficient method (Ministry of Land, Infrastructure, and Transport, 2018). In this study, the skin friction of the pile foundation was calculated by applying the α coefficient method under the undrained condition.

Since the α coefficient method is applied when the pile is embedded in the saturated clay layer, it is $\phi_{u}=0$ in the undrained condition, thus δ=0. At this time, the skin friction of the pile is expressed as the adhesion force between the pile and the ground.
(3)$$f_{s}=c_{s}=\alpha c_{u}$$
where $c_{s}$ is the cohesion force (kN/m^2^); α is the cohesion force coefficient; and $c_{u}$ is the cohesion force in undrained conditions (kN/m^2^).

The α value varies depending on the hardness of the clay layer, the type and construction method of the pile, the stratum condition, and the size of the pile.

### 2.3. C.T Cofferdam Method

The C.T cofferdam method creates a cut-off space using prefabricated steel caisson and rubber materials when installing, repairing, or reinforcing underwater structures.

There are rubber tubes and compression-type rubber packing materials used for the cut-off. As shown in Figure 2a, the adhesive rubber tube attached to the end of the steel caisson is in close contact with the leveling concrete and creates a pressure greater than the water pressure to block the external inflow. As shown in Figure 2b, the compression-type rubber packing compresses the rubber packing attached to the bottom of the steel caisson with the caisson’s own weight, which is greater than the water pressure, to block the external inflow to the leveling concrete. In this study, a close-fitting type rubber tube C.T cofferdam was used.

## 3. Review of Design Data

### 3.1. Site Overview

In the case of OO Bridge P2, the pre-bored pile was pulled out by about 0.17 to 0.34 m due to the buoyancy generated at the bottom of the leveling concrete and the C.T cofferdam was levitated, causing damage to the structure.

Figure 3 shows the pull-out of the C.T cofferdam at P2 where the problem occurred.

### 3.2. Analysis of Pull-Out Force of Pre-Bored Piles According to Design Criteria

With reference to various domestic and foreign design standards, the pull-out forces of pre-bored piles and drilled shafts embedded in rock were calculated and compared. When calculating the design structure of the C.T cofferdam, the pull-out force of the pile applied was 600 kN. It was checked whether this pull-out force satisfies the design criteria, with the theoretical equation substituting the actual insertion depth of the pile (about 1.92 m).

(1)C.T Cofferdam Design Structural Calculation Results

As shown in Table 2, the total weight of C.T cofferdam was 3665.250 kN, the water load inside the tank was 5078.34 kN, the water load inside the watertight part was 548.311 kN, and the weight of the leveling concrete was 7591.710 kN. When calculating the load in the design structural calculation sheet, the pile resistance was 600 kN (at the time of the design) and the stability of the buoyancy was calculated to be 1.4. Since it is higher than 1.2, the buoyancy was considered stable.

(2)Review of the theoretical equation for calculating the pull-out force of a pre-bored pile according to the design criteria

Since most of the pull-out force of pre-bored piles is generated from the skin friction of the piles embedded in the bedrock (excluding the sedimentary section at the top of the bedrock), the skin friction of the piles embedded in the bedrock according to each design standard was calculated.

Considering no design standards for estimating the skin friction of pre-bored piles embedded in bedrock have been presented domestically and abroad, referring to the drilled shaft design standards, the skin friction of the bedrock was calculated.

To examine the skin friction of main pile, (1) Korean Geotechnical Engineering [23], Ministry of Land, Infrastructure, and Transport [24], (2) Cho [25], and (3) Cho [26] were referred to. As the condition to be reviewed was similar to that of the anchor method, the study of Kim and Jung [27] was also referred to for review. As shown in Table 3, as a result of examining the grouting construction data, the grouting depth was found to be about 1.92 m on average and the skin friction value was calculated based on that value.

### 3.3. Adhesive Force Analysis of Pile and Grout

#### 3.3.1. Adhesive Force Concept of Pile and Grout

The skin friction failure of the pre-bored pile embedded in the bedrock appears in four forms, as shown in Figure 4. These are the shear failures between the pile-filling material and the pile; the adhesion failure between the rock and the pile-filling material; the shear failure of the rock mass itself; and finally the shear failure of the pile-filling material itself. Where the shear failure with the least resistance occurs, it leads to skin friction failure.

Since the uniaxial compressive strength of most rocks is generally higher than the uniaxial compressive strength of the grout, shear failure usually occurs in the pile-filling material before the shear failure of the rock and the skin friction of the pile embedded in the rock is determined by the adhesion strength between the pile and the pile-filling material.

Moon et al. [12] studied the effect of the W/C (water/cement, hereinafter called W/C) ratio of the pile-filling material and the fine aggregate mixing ratio on the uniaxial compressive strength, and the uniaxial compressive strength of the pile-filling material was about 7–36 MPa depending on the mixing ratio. Moon et al. [11] measured the load and displacement at the pile head after injecting grout with a W/C ratio of 50% into basalt perforated with a diameter of BX (BX is a casing standard of out diameter of 59.0 mm used for standard penetration tests) and inserted a structural steel pipe (outer diameter of 48.6 mm). The skin friction of the steel pipe pile embedded in the bedrock was determined by the adhesion strength of the steel pipe and the grout, and the unit ultimate skin friction was in the range of 0.92 to 1.04 MPa.

In a study by Moon and Park [13], in the absence of fine aggregate, the unit ultimate skin friction was 0.54 to 0.56 MPa regardless of W/C. Table 4 shows our field conditions (D = 0.6 m in diameter and 1.92 m in rock penetration condition) using the unit ultimate skin friction values determined in a previous study by Moon et al. [11] and Moon and Park [13]. As shown in Table 4, this is the result of calculating the allowable skin friction. The allowable skin friction was about 651.1 to 1109.29 kN.

#### 3.3.2. Lab Test for Measuring Adhesion Strength

The skin friction of the pre-bored pile embedded in the rock should be evaluated by the adhesion and skin friction occurring on the surface of the grout and pile foundation, rather than those of the ground and grout. In addition, during on-site construction, it should be considered that the adhesion is reduced due to seawater inflow in the drilling hole, slurry, and a poor W/C ratio. Therefore, in order to evaluate the skin friction of the pre-bored pile, five reinforced concrete pile samples collected at the work site were used. In using this pile, the grout is poured underwater at 70% of the W/C ratio and the adhesive strength of the pile foundation and grout in the bedrock was inspected for 28 days by air curing.

(1)Blending process

In order to apply a mixing ratio similar to the field conditions, the W/C ratio was set to 70% and grout was poured on the surface of the pile foundation, collected from the field under both air and underwater conditions, as shown in Figure 5. Similarly, a test piece was prepared.

Table 5 shows the concrete mixing ratio for the lab pull-out test.

(2)Test process

The lab test was conducted to confirm the adhesion strength of the surface of the pile foundation and grout. As shown in Figure 6a, the surface of the grout was cut into 4 cm × 4 cm sizes and the epoxy was coated on the surface of the pull-out equipment; the grout was as shown in Figure 6b. As shown in Figure 6c, the pull-out force was tested to evaluate the adhesion strength. Figure 6d is a view after the test.

(3)Test results

The adhesion strength of the three points of the air curing of concrete was in the range of 0.65–0.81 MPa and the adhesion of the two points of the water curing of concrete was in the range of 0.43–0.47 MPa. Table 6 shows the adhesion strength of the tensile member, the grout, and the calculation results of the allowable skin friction of the test piece.

Even though the experimental methods are different, the test results applying air curing of the concrete are similar to those of previous studies, thus the results are considered reliable. In the case of the water curing of concrete, it represents 60% of the adhesion strength of the air curing of concrete and when considering various variables (slurry, seawater inflow, etc.) occurring in the field, the adhesion strength is expected to be lower.

### 3.4. Results of Pull-Out Load Test

There are many design standards for designing C.T cofferdams with a pull-out force of 600 kN or more, but most of these standards are for drilled shafts. Since this study used the design standard of pre-bored piles based on empirical equations that have not yet been clearly established, various reviews are needed.

In general, the pile load test for the bearing capacity test was performed according to the standard after the pile construction was completed and the dynamic load or static load test was performed within 5% of the number of piles. However, since there was no similar guideline in the pull-out load test, the pull-out force for the pre-bored pile at the point was measured to examine the pull-out problem that occurred at the research site.

The most accurate measurement of pull-out force was 217 kN in the pull-out load test performed on the T1 pile of P2 at the OO Bridge in April 2018. However, this was tested 13 days after the grout was placed and 78.7% of the measured value was set as the allowable pull-out force by referring to the Basic Structure Design Standards and Explanations, Road Bridge Design Standards, and other references (Table 7).

The correction value was calculated by referring to the ACI 208R-92 standard and estimating the uniaxial compressive strength according to the material age. The on-site ultimate pull-out test value was 629.2 MPa and the safety side’s age of 12 days was applied in consideration of the curing time and pull-out test time. Therefore, the correction factor according to the age of the concrete for 12 days was 0.828 and here, again, 5% was corrected for the water curing of concrete; the final correction factor was 0.787. The calculated pull-out force was 629.2 MPa/0.787 = 799.95 MPa and the short-term load safety factor of 2 was applied here; the final pull-out force was calculated as 399.7 kN. The allowable pull-out force reviewed based on the on-site pull-out load test was 399.7 kN, which is very different from the pull-out force of 600 kN applied to the design. It is agreed that poor W/C ratio, due to the inflow of seawater, slurry, and excessive inflow of seawater into the drilling hole at the site, is the cause of lowering the adhesion between the pile and the grout.

A total of 399.7 kN from the pull-out load test is about 60% of the allowable pull-out force f 600 kN applied to the design and therefore it is agreed that the pre-bored pile was pulled out due to the lack of the design load. In addition, in comparing the results of the pull-out load test with the theoretical results, the results are most similar to those of CGS (1985). This study compared the uniaxial compressive strength of rock and concrete, which is similar to the concept of estimating the skin friction of a pre-bored pile embedded in bedrock, thus it seems to show similar results to our experiment. 

### 3.5. C.T Cofferdam Stability Review

In order to estimate the skin friction of the pre-bored pile embedded in the rock, the original design, analysis of the research data, review of the design criteria, adhesion strength test of the pile foundation and grout, the results of the field pull-out load test were reviewed. Based on this, the stability of C.T cofferdam was reviewed by recalculating the pull-out force similar to when calculating to the field conditions. As a result of examining the stability of C.T cofferdam as shown in Table 8, the minimum safety factor is satisfied when the pull-out force of the pile foundation is at least 500 kN. 

While the pull-out force of the original design was designed based on 600 kN, as a result of the on-site experiment test, it was found to be unstable at 399 kN. In addition, when reviewed by various design standards, it was unstable at 220–1700 kN. Therefore, considering various variables (slurry, seawater inflow, etc.) occurring at the site, the adhesion strength can be further lowered, thus the depth of penetration of the rock and the W/C ratio should be sufficiently considered when designing.

## 4. Reviewing the Amount of Pull-Out of Pre-Bored Piles through Numerical Analysis

### 4.1. Analysis Conditions

In order to identify the amount of pull-out of the pile foundation, depending on leveling the concrete base buoyancy, the shape of the pile and field ground conditions were considered, and soil properties of the field were estimated based on the site investigation reports and empirical equations. 

In this study, two boring investigations were conducted. The sites are composed of sedimentary layers, weathering soils, weathered rocks, and soft rock layers from the top, and the ground water table was GL(-) 8.3 to 8.5 m in the upper plate of the temporary bridge, being distributed in sedimentary layers. Therefore, this ground condition was modeled for numerical analysis. 

#### 4.1.1. Analysis Program

In this study, Midas GTS (Midas IT. Co.,Ltd., Gyeonggido, Korea), a finite element analysis program, was employed. Midas GTS can perform modeling to consider the ground shape and construction process as much as possible during ground analysis, and can reflect actual field conditions in the light of non-linearity and in-situ ground stress states of various materials used for numerical analysis. 

For analyzing complex non-linear behaviors of the ground, Midas GTS was equipped with various constitutive elasto-plastic models, as well as with automatic mesh generation. 

#### 4.1.2. Drilling Section

The ground on the upper plate was investigated from the 270° direction to the 90° direction of C.T cofferdam to collect the ground data to be applied to the model. Sedimentary layers were confirmed in the current surface, the thickness was distributed as 12.2–16.3 m, and the composition consisted of neutral and fine yarns. The N value measured in the standard penetration test was 1/30 to 12/30 (times/cm) and the relative density was in a very loose to moderately dense state. The weathered soil layer was distributed in the BH-1 hole with a thickness of 1.3 m, the composition was of silt sand, and the condition was dense. The weathered rock layer was in a state of severe weathering and was distributed at a thickness of 4.8–6.4 m, wherein the composition was of silt sand. As a result of the standard penetration test, the N value was 50/10 to 50/3 (times/cm) and the relative density was very dense. The soft rock layer was distributed at a thickness of 2.0 m in the entire borehole and it was observed with the naked eye that the rock condition was severely to moderate weathering. In addition, the rock strength was weak to moderately strong.

Figure 7 is a stratum section view used for the analysis section.

#### 4.1.3. Calculation of the Material Properties of the Ground

The Mohr–Coulomb (MC) model was used for the natural ground and the composite pile was modeled both by applying the steel pipe pile SKK490 and by the reinforced concrete pile specification. The geological structure of the site was composed of a sedimentary layer, weathered soil layer, weathered rock layer, and soft rock layer, and its structure was relatively simple; very soft sand was distributed in the upper original ground.

In order to determine the design constant of the ground, the characteristic values for the cohesion force, internal friction angle, and elastic modulus of each stratum were calculated based on the ground survey report, empirical equations, and other references.

Table 9 and Table 10 show the ground constants used for numerical analysis. The material properties applied to the grout and injected into weathered rock and soft rock were selected in full consideration of the site conditions. The strength of 399.7 kN in the pull-out load test was about 45.4% of the strength of 881 kN in the air curing of concrete, thus 40% was used for safety reasons. Therefore, 928,000 kN/m^2^, which is 40% of the concrete elastic modulus of 2,320,000 kN/m^2^, was determined as the elastic modulus of the grout, considering the effects of seawater and slime, and was applied to the numerical analysis. The modulus of elasticity is a factor that greatly affects the amount of elastic settlement and the modulus of elasticity of weathered rocks and soft rocks is not the same. The modulus of elasticity of the grout was applied to both the weathered rock and the soft rock in consideration of the field conditions after the application of the grout.

#### 4.1.4. Modeling

The internally excavated composite pile modeled was composed of a steel pipe pile with a diameter of 600 mm and thickness of 14 mm from the leveling concrete to a depth of 6.5 m, which was a reinforced concrete pile with a diameter of 600 mm from the end of the steel pipe pile to the tip. The stratum structure was modeled as shown in Figure 8 based on the BH-2 drilling column.

In the numerical analysis, the composite steel pipe pile and reinforced concrete pile were modelled as elastic materials and a leveling concrete base buoyancy of 29,929.973 kN/m^2^ was applied to the upper leveling concrete upwards.

Figure 9 and Figure 10 present the analysis results and show the maximum value of the vertical displacement in each node. The pile end had the maximum displacement in the nodes, as referred to in the boxes in Figure 9 and Figure 10.

For the modeling of pre-bored piles embedded in the rock, material properties of weathered rock and soft rock, where the piles are embedded, were compared on the basis of the material properties of grout materials with a low elastic modulus, as shown Table 9.

### 4.2. Numerical Analysis Review

In the case of the P2 lateral direction (90°→270°), as shown in Figure 9a, the pull-out displacement of the tip of the pile foundation was 0.22 to 0.29 cm and the total pull-out displacement of the structure was 0.70 to 0.77 cm during the integral movement, as shown in Figure 9a. It is within safe limits because it is smaller than the safety standard of 1 inch (2.54 cm). Figure 9b is the result of the on-site pull-out load test and the numerical analysis reflects the field conditions. For the modulus of the grout injected in the weathered rock and soft rock, 40% of the modulus of the elasticity of concrete was applied. As a result of the numerical analysis, the pull-out displacement at the tip of the pile foundation was 0.64 to 0.89 cm and the overall pull-out displacement was 2.37 to 2.61 cm. It was higher than 1 inch (2.54 cm) and outside of safety standards.

In the case of the direction perpendicular (0°→180°) to the P2 pier, as shown in Figure 10a, the pull-out displacement of the tip of the pile foundation was 0.22 to 0.29 cm and the overall pull-out displacement was 0.70 to 0.77 cm, which is within the safe range. However, as depicted in Figure 10b, considering the field conditions, it was found that the total pull-out displacement of the structure was 2.37 to 2.61 cm, similar to the lateral direction and thus unstable.

As a result of the numerical analysis, the integral behavior of the pile foundation and the ground appeared to be stable when the site conditions were not sufficiently considered, but in the numerical analysis considering the site conditions, it was found that the pull-out displacement exceeded the standard of 1 inch and was unstable. Therefore, when designing the skin friction (pull-out force) of the pre-bored pile embedded in bedrock, the site conditions should be sufficiently considered.

Table 11 summarizes the numerical analysis results.

## 5. Conclusions

In this study, the pre-bored pile of OO bridge pier P2 and the pull-out phenomenon observed at C.T cofferdam were reviewed according to the design standards of the pre-bored pile reflected in the design. The stability of C.T cofferdam was reviewed by evaluating the adhesion strength of the grout and concrete surface attached to the bedrock. The following conclusions were drawn.

According to the existing design data and the theoretical equation for skin friction embedded in bedrock, most of the allowable skin friction forces were more than 600 kN, thus it was safe in design. However, in the field, the pre-bored pile showed instability. This means that the skin friction of the drilled shaft applied during the design and the pre-bored pile applied during the construction were not the same, and it seems to have occurred because the pull-out resistance of the pre-bored pile was the lower one of the drilled shafts.

As a result of the lab test for adhesion strength, the strength during the air curing of concrete was overestimated as 881 kN compared to the pull-out force of 600 kN as applied during the design and the strength for the water curing of concrete was 542 kN, which showed similar results. However, when various field variables (slurry, seawater inflow, etc.) occurring in the field were taken into consideration, the adhesion strength was expected to be lower.

As a result of re-examining the allowable pull-out force according to the pull-out load test, the pull-out force is 399.7 kN, which is about 66% of the pull-out force of 600 kN applied to the design. The results of the on-site pull-out load test and the theoretical equation were compared. The method of obtaining the skin friction of the pre-bored pile and the method of obtaining both the uniaxial compressive strength of the rock and the concrete pile were similar, and the skin friction calculated by these two methods showed similar results as well.

As a result of the numerical analysis review, in the case of the integral pull-out behavior of the pile foundation and ground, the displacement of the entire structure was stable within the allowable value, but when using the material properties of the grout, considering the field conditions, the displacement of the entire structure exceeded the allowable value. This appears to have lowered the adhesion strength due to construction issues such as ground complexity as well as seawater and slime treatment, which were not expected at the time of design.

## Figures and Tables

**Figure 1 materials-14-05593-f001:**
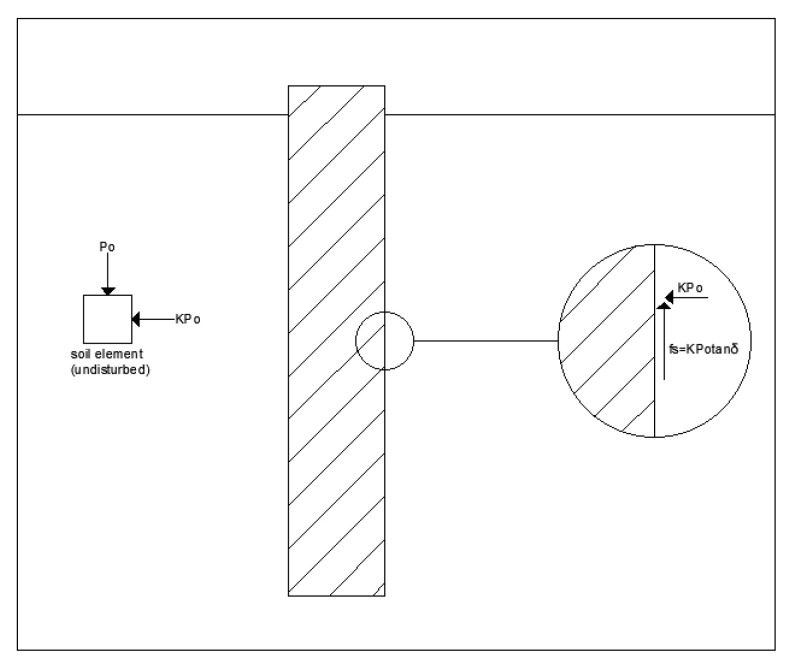
Skin friction force on the pile.

**Figure 2 materials-14-05593-f002:**
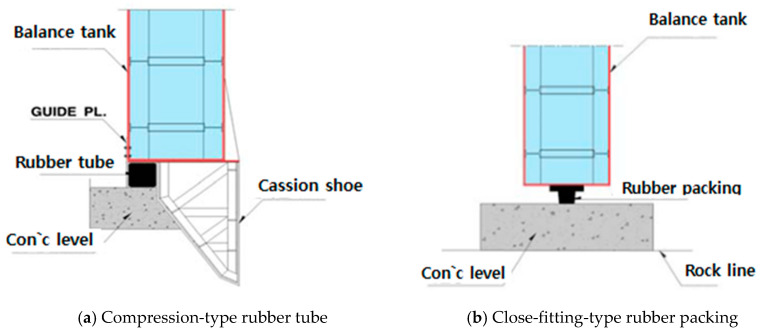
Schematic of C.T cofferdam method.

**Figure 3 materials-14-05593-f003:**
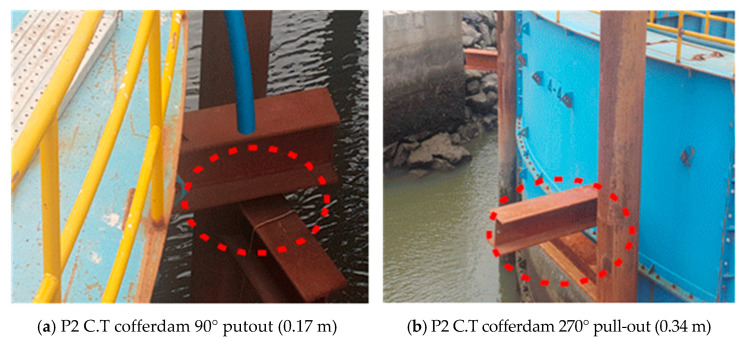
Pull-out view of C.T cofferdam at P2.

**Figure 4 materials-14-05593-f004:**
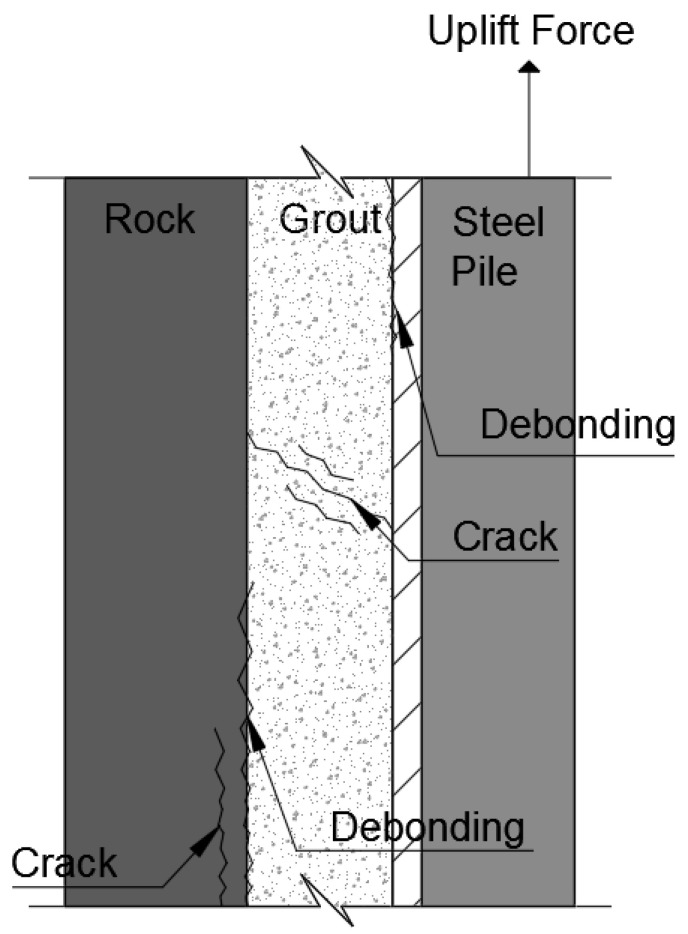
Pull-out fracture modeling of pre-bored piles.

**Figure 5 materials-14-05593-f005:**
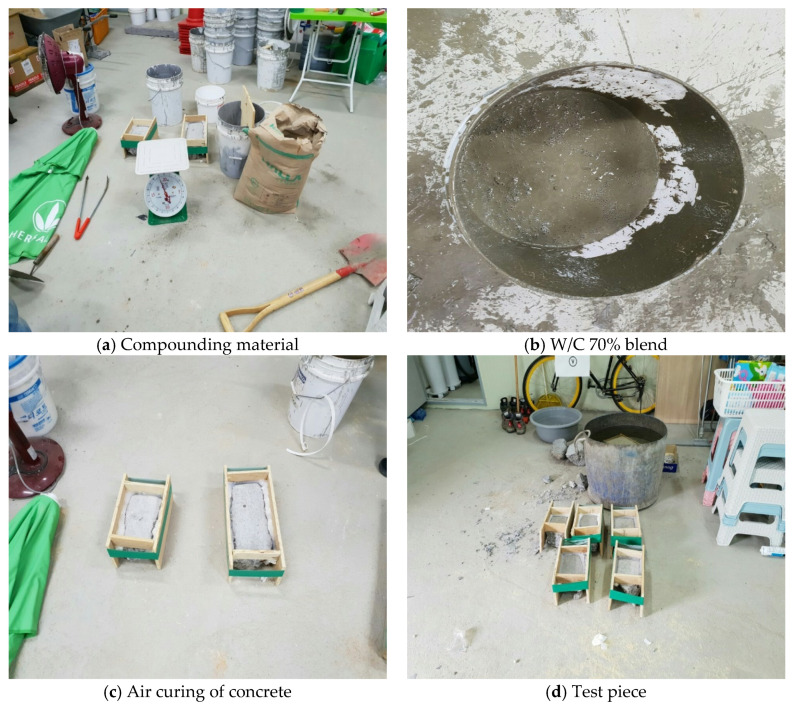
Test piece production process.

**Figure 6 materials-14-05593-f006:**
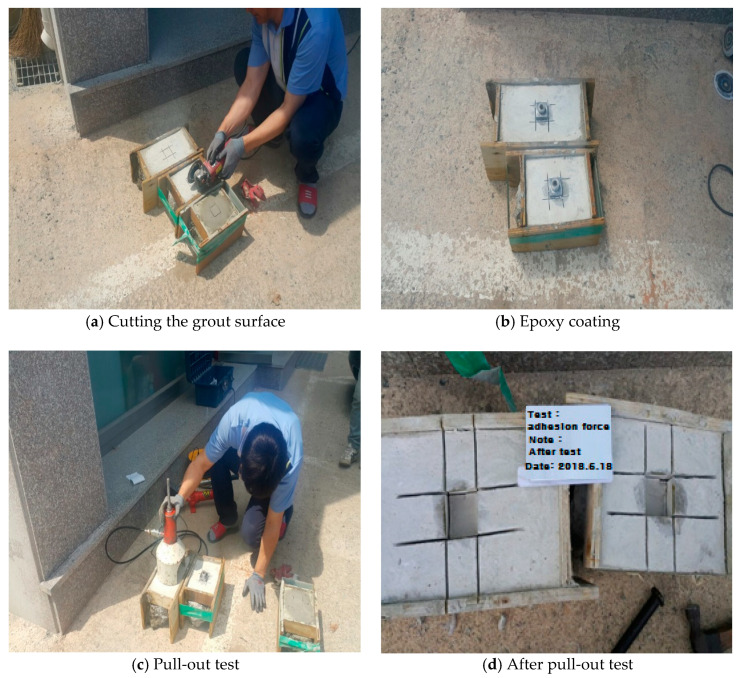
A view of the test to check the adhesion force.

**Figure 7 materials-14-05593-f007:**
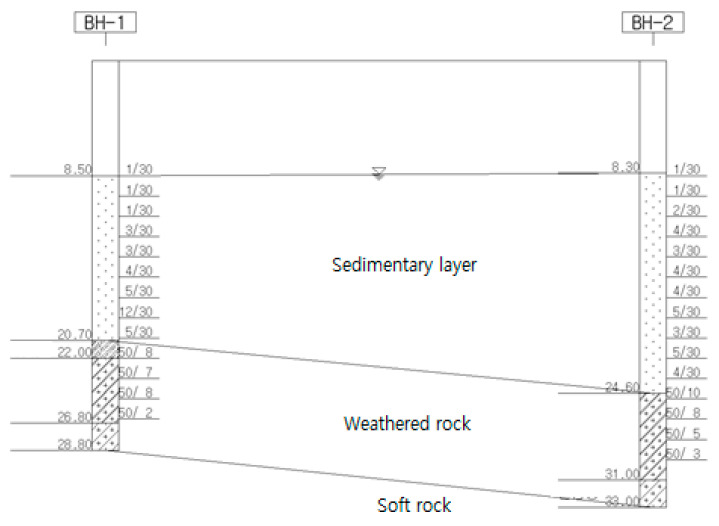
Stratum section view.

**Figure 8 materials-14-05593-f008:**
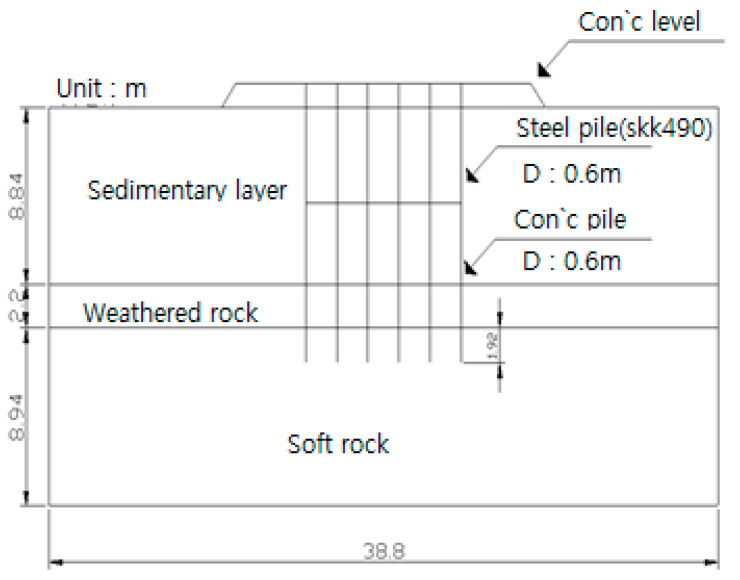
Analysis section.

**Figure 9 materials-14-05593-f009:**
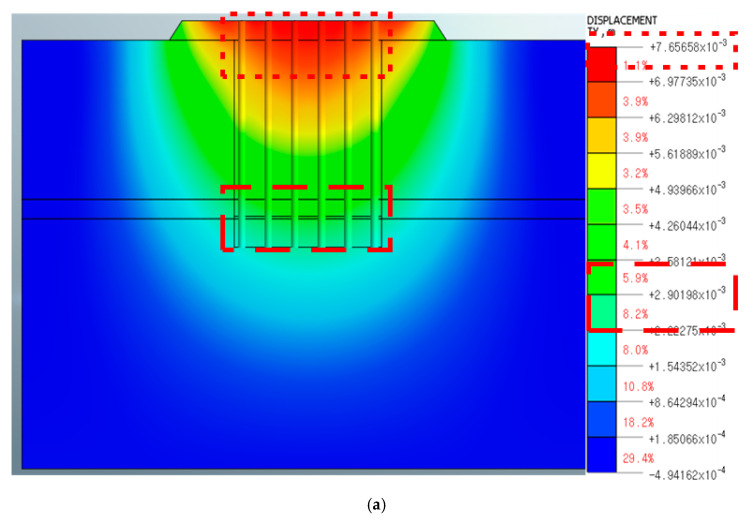
Numerical analysis result in the P2 direction, (**a**) Soft rock condition at 1.92 m and the integrated behavior of both the pile foundation and ground; (**b**) The soft rock insertion condition of 1.92 m and the on-site pull-out load test results are reflective of this.

**Figure 10 materials-14-05593-f010:**
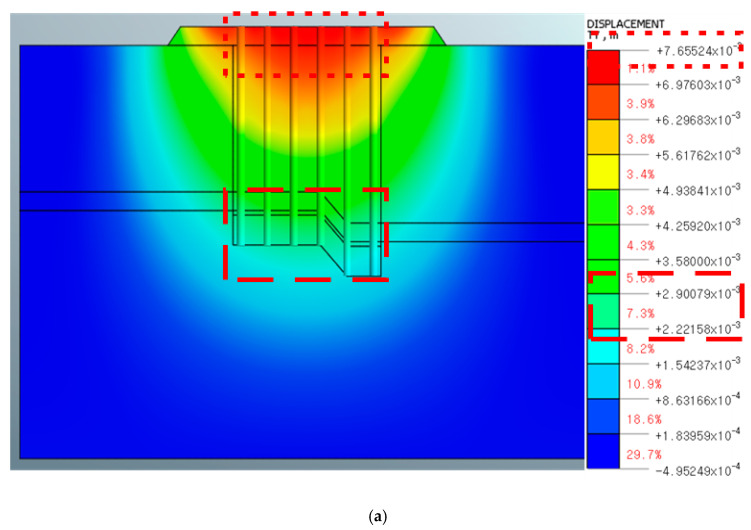
Numerical analysis results in the perpendicular direction of the P2 pier. (**a**) Soft rock condition at 2.21 m and the integrated behavior of both the pile foundation and ground. (**b**) Soft rock penetration condition at 2.21 m, reflecting the results of the on-site pull-out load test.

**Table 1 materials-14-05593-t001:** Physical properties of reinforcements used for tests.

Classification	Equations	Proposers	Applicability
Skin Friction	$f_{s}=a_{c}+q`Ktan\delta$	Tomlinson (1972)	Pre-stress and clay soil
	$f_{s}=K{`}_{q}tan\delta =\beta q`$	Jennings and Burland (1973)	Effective stress amnd clay soil
	$f_{s}=\lambda \left(q`+2s_{u}\right)$	McClelland and Focht (1967)	Mixing method and clay soil
	$f_{s}=a_{cu}$	Meyerhof (1976)	Clay soil
	$f_{s}=K_{s}\sigma {`}_{v}tan\delta$	Meyerhof, Colye-Castello, Vesic, API etc	Sandy soil
End-bearing capacity	$q_{p}=p_{0}N_{q}\;\leq 5N_{q}tan\phi$	Meyerhof (1976)	Sandy soil
	$q_{p}=cN_{c}$	Meyerhof (1976)	Clay soil
	$q_{p}=cN_{c}+\sigma_{0}N_{0}$	Vesic (1977)	Cavity expansion theory
	$q_{p}=A_{k}\gamma B+B_{k}q_{T}\;\left(q_{T}=a_{T}\gamma D\right)$	Berezantzev (1961)	

**Table 2 materials-14-05593-t002:** Stability review for the buoyancy of C.T cofferdam.

Classification		Selected Load (kN)
Resistance Load	C.T cofferdam weight	3665.250
	Water load inside the tank	5078.340
	Water load inside watertight part	548.311
	Leveling concrete weight	7591.710
	Resistance of piles (600 kN per pile)	21,600.000
	Pile weight (45.2 kN per pile)	1630.000
Buoyancy	Leveling concrete bottom buoyancy	29,929.973
Fs (Satety rate)	$\mathrm{FS}=\frac{3665.25+5078.34+548.311+7591.71+21,600}{29,929.973}=1.4>1.2$	

**Table 3 materials-14-05593-t003:** Comparison of the pull-out force applied to the design and design standard values.

Classification		Allowable Skin Friction of Pile (pull-out, kN)	Remark
Existing design data	C.T cofferdam design structure calculation sheet (1.0 m embedded in the rock)	600	Depth embedded in the rock 1.92 m Diameter of the pile 0.6 m
	Pull-out force considering the thread-embedded depth of the pile (2.197 m embedded in the rock)	1209	
Design standard	AASHTO (1996)	200.5	
	Canadian Geotechnical Society (1985)	356.9	
	NAVFAC (1982)	931.8	
	FHWA (1988) Horvath and Kenny (1979)	648.8	
	FHWA (1999) Horvath and Kenny (1979)	770.4	
	FHWA (1999) Rowe and Armitage (1984)	746.8	
Structural Foundation Design Standard (2008)	Wiliams et al. (1980)	659.6	
	Rowe and Armitage (1987)	1774,3	
	Horvath and Kenney (1979)	819.9	
	Carter and Kulhawy (1988)	770.9	
	Reynolds and Kaderabek (1987)	3460.5	
	Gupton and Logan (1984)	2307.8	
	Reese and O‘Neil (1987)	1730.8	
	Rosengerg and Journaeaux (1976)	1489.7	
Anchor method	Putout resistance of anchor	292.8	
	Friction of ground and grout	566.7	
	Maximum adhesive force of tensile material and grout	791.1	

**Table 4 materials-14-05593-t004:** Allowable skin friction between tension member and grout.

Classification	Allowable Skin Friction	Test Methods
Moon et al. [11]	$Q_{u}=f_{s}\times \pi \times D\times L\;Q_{u}=920\times \pi \times 0.6\times 1.92\;=3327.89\;\mathrm{kN}\;Q_{a}=\frac{Q_{u}}{FS}=\frac{3327.89}{3}=1109.29\;kN$ Test conditions: W/C: 50%	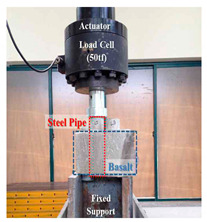
Moon and Park [13]	$Q_{u}=f_{s}\times \pi \times D\times L\;Q_{u}=540\times \pi \times 0.6\times 1.92\;=1953.33\;\mathrm{kN}\;Q_{a}=\frac{Q_{u}}{FS}=\frac{1953.33}{3}=651.11\;kN$ Test conditions: W/C: 60~120%	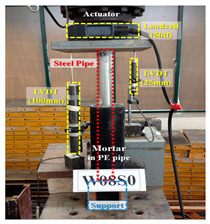

**Table 5 materials-14-05593-t005:** Mixing ratio for lab pull-out test.

Classification	Water (mL)	Cement (g)	Grout Material (g)
Water curing of concrete: 28 days, 2 EA	490	700	70
Air curing of concrete: 28 days, 3 EA	490	700	70

**Table 6 materials-14-05593-t006:** Allowable skin friction according to the adhesion between the pile and grout.

Classification		Allowable Skin Friction	Avgerage Allowable Skin Friction
Water curing	WT-1	$Q_{u}=f_{s}\times \pi \times D\times L\;Q_{u}=430\times \pi \times 0.6\times 1.92\;=1555.4\;\mathrm{kN}\;Q_{a}=\frac{Q_{u}}{FS}=\frac{1555.4}{3}=518.4\;\mathrm{kN}$	542.55 kN
	WT-2	$Q_{u}=f_{s}\times \pi \times D\times L\;Q_{u}=470\times \pi \times 0.6\times 1.92\;=1700.12\;\mathrm{kN}\;Q_{a}=\frac{Q_{u}}{FS}=\frac{1700.12}{3}=566.7\;\mathrm{kN}$	
Air Curing	AT-1	$Q_{u}=f_{s}\times \pi \times D\times L\;Q_{u}=810\times \pi \times 0.6\times 1.92\;=2929.99\;\mathrm{kN}\;Q_{a}=\frac{Q_{u}}{FS}=\frac{2929.99}{3}=976.66\;\mathrm{kN}$	881.03 kN
	AT-2	$Q_{u}=f_{s}\times \pi \times D\times L\;Q_{u}=730\times \pi \times 0.6\times 1.92\;=2640.61\;\mathrm{kN}\;Q_{a}=\frac{Q_{u}}{FS}=\frac{2640.61}{3}=880.2\;\mathrm{kN}$	
	AT-3	$Q_{u}=f_{s}\times \pi \times D\times L\;Q_{u}=650\times \pi \times 0.6\times 1.92\;=2358.72\;\mathrm{kN}\;Q_{a}=\frac{Q_{u}}{FS}=\frac{2358.72}{3}=786.24\;\mathrm{kN}$	

**Table 7 materials-14-05593-t007:** Re-examination of allowable pull-out force according to the results of the pull-out load test.

Classification		Re-examination of Allowable Pull-Out Force	Allowable Pull-Out Force
On-site pull-out test	P2	① $Q_{u}=651.1\;\mathrm{kN}$ (at Offset Point) $Q_{u}=651.1÷0.787=827.3\;\mathrm{kN}Q_{u}=827.3÷2=413.6\;\mathrm{kN}$ ② $Q_{u}=629.2\;\mathrm{kN}$ (at 0.25 in) $Q_{u}=629.2÷0.787=799.4\;\mathrm{kN}\;Q_{u}=799.4÷2=399.7\;\mathrm{kN}$	399.7 kN

**Table 8 materials-14-05593-t008:** Results of C.T cofferdam stability review according to the recalculation of the pull-out force.

Classification			Pile Pull-Out Force (kN)	Safety Factor (Standard 1.2)		Remark
P2	Existing design	Original design (1.0 m rock-bottom embedded)	600	1.4	stable	Estimated value
	Lab experiment	Grout air curing of concrete (1.92 m rock depth)	881	1.7	stable	Site conditions lacking
		Grout placed in water (1.92 m rock depth)	542	1.3	stable	Only the conditions for placing it in water were taken into account
	On-site experiment	On-site pull-out test (1.92 m rock depth)	399	1.1	instable	Test methods have high reliability
	Design data	Designer review (2.197 m rock depth)	1210	2.0	stable	Cast-in-place pile standards
	Previous research	Various research standards (1.92 m rock depth)	680~1160	1.4~2.0	stable	Air curing of concrete
		Various design standards (1.92 m rock depth)	220~1700	0.8~2.6	stable~ instable	Cast-in-place pile standards

**Table 9 materials-14-05593-t009:** Material properties applied to the numerical analysis.

Classification		Sedimentary Soil	Weathered Rock	Soft Rock	Leveling Concrete	Rebar Concrete	Steel Pipe Pile
Wetting unit weight	$\gamma_{t}$ (kN/m3)	17	23	25	25	25	78
Saturation unit weight	$\gamma_{sat}$ (kN/m^3^)	18	23.5	25.5	25.5	25.5	79
Cohesion	c (kN/m^2^)	13	30	50	-	-	-
Internal friction angle	ϕ (°)	0	33	35	-	-	-
Elastic modulus coefficient	E (kN/m^2^)	4000	167,500	2,180,000	2,320,000	2,320,000	2.1 × 10^8^
Poisson’s ratio	ν	0.38	0.3	0.26	0.2	0.19	0.3

**Table 10 materials-14-05593-t010:** Applied material properties of the grout injected into the weathered rock and soft rock.

Classification		Weathered Rock (Injection Material)	Soft Rock (Injection Material)
Wetting unit weight	$\gamma_{t}$ (kN/m^3^)	23	25
Saturation unit weight	$\gamma_{sat}$ (kN/m^3^)	23.5	25.5
Cohesion	c (kN/m^2^)	30	50
Internal friction angle	ϕ (°)	33	35
Elastic modulus coefficient	E (kN/m^2^)	928,000	928,000
Poisson’s ratio	ν	0.3	0.3

**Table 11 materials-14-05593-t011:** Numerical analysis results.

Classification		Displacement (cm)	Stability (Based on 1 inch)
P2 lateral direction Soft rock at 1.92 m	Pile tip	0.22 to 0.29	0.29 ≤ 2.54_stable
	Total displacement	0.70 to 0.77	0.77 ≤ 2.54_stable
P2 lateral direction Underwater grout Consider material properties	Pile tip	0.64 to 0.89	0.89 ≤ 2.54_stable
	Total displacement	2.37~2.61	2.61 ≥ 2.54_unstable
P2 bridge direction Soft rock at 2.21 m	Pile tip	0.22 to 0.29	0.29 ≤ 2.54_stable
	Total displacement	0.70 to 0.77	0.77 ≤ 2.54_stable
P2 bridge direction Underwater grout Consider material properties	Pile tip	0.64 to 0.89	0.89 ≤ 2.54_stable
	Total displacement	2.37~2.61	2.61 ≥ 2.54_unstable

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
