# Peer review of "Evaluation of the Pullout Behavior of Pre-Bored Piles Embedded in Rock"

_materials, 2021, doi:10.3390/ma14195593_

Round 1

Reviewer 1 Report

The manuscript attempts a study to evaluate the pull out behavior of pre-bored pile embedded in rock using lab-scale experiments and numerical investigations. The study presented in this paper is interesting and perhaps valuable addition to the existing knowledge of foundation engineering and offshore structures. Nonetheless, in its current state the manuscript presents deficiency in a well-structured writing, has various errors and few missing technical discussion to the best of my knowledge. Some of these are highlighted in my comments below. More importantly, the manuscript requires careful review of the English language and connectivity in presenting the discussions. To this end, I feel that the content of the manuscript has potential in contributing to the technical knowledge, however requires a thorough revision. It is for this reason; I suggest that the manuscript be sent back to the authors for major revision followed by another review.

Below are some of the comments/suggestions noted during my review. I believe these comments/suggestions will assistance the authors in revising the manuscript as deemed appropriate.

Figures: I see that authors have directly copied few figure/illustration from other papers (for example Figure 1, Figure 2, Figure 4). Though the figures are duly cited, I suggest the authors and editorial office to check is copy write clearance is required for these figures.

Lines 37 and 41: The authors here refer to “..accurate calculation equation..” and “..equation for calculating the skin friction..”; I believe, this should be written as “codal provisions to calculate…”.

Line 78: What is the full form on “W/C”? When a short form is used for the first time in the manuscript, please mention the full form for the readers to understand. I understand that most of the engineers working in concrete will recognise this, but you need to be fair to all the readers interested in your manuscript.

Line 79-80: What is the full form on “(t-z)”? And what do you mean by N-Values. Please mention the full form and define these terms clearly for the readers to understand. For example, ‘N-Value’ should be ‘N-Value obtained from the Standard Penetration Test for the starta…’

Line 165: The section “Site Overview” should be elaborated a bit more. In the current state the section does not really convey any description of the site, expect the visuals in Figure 3.

Line 215: What is the full form on “BX”? When a short form is used for the first time in the manuscript, please mention the full form for the readers to understand.

Table 7: The results presented in this table is the key information in this manuscript. However, it is fair to tabulate the result in a table and expect the reader to understand them completely. Authors SHOULD present a well structures discussion of the results and preferably compare with exiting research.

Section 4: Authors do not explain the software program and procedure used in the numerical analysis. This should be clearly mentioned and discussed. The accuracy of the results depends on various steps used in numerical modelling, this is very important part. This SHOULD be revised.

Author Response

Response to reviewer’s comment

Thank you so much for the reviewer’s invaluable comments on the paper. We revised the paper according to reviewer’s comments.

Reviewer 1

Comments and Suggestions for Authors

The manuscript attempts a study to evaluate the pull out behavior of pre-bored pile embedded in rock using lab-scale experiments and numerical investigations. The study presented in this paper is interesting and perhaps valuable addition to the existing knowledge of foundation engineering and offshore structures. Nonetheless, in its current state the manuscript presents deficiency in a well-structured writing, has various errors and few missing technical discussion to the best of my knowledge. Some of these are highlighted in my comments below. More importantly, the manuscript requires careful review of the English language and connectivity in presenting the discussions. To this end, I feel that the content of the manuscript has potential in contributing to the technical knowledge, however requires a thorough revision. It is for this reason; I suggest that the manuscript be sent back to the authors for major revision followed by another review.

Below are some of the comments/suggestions noted during my review. I believe these comments/suggestions will assistance the authors in revising the manuscript as deemed appropriate.

Figures: I see that authors have directly copied few figure/illustration from other papers (for example Figure 1, Figure 2, Figure 4). Though the figures are duly cited, I suggest the authors and editorial office to check is copy write clearance is required for these figures.

Response: This has been corrected.

Lines 37 and 41: The authors here refer to “..accurate calculation equation..” and “..equation for calculating the skin friction..”; I believe, this should be written as “codal provisions to calculate…”.

Response: It is mentioned in Table 1.

Line 78: What is the full form on “W/C”? When a short form is used for the first time in the manuscript, please mention the full form for the readers to understand. I understand that most of the engineers working in concrete will recognise this, but you need to be fair to all the readers interested in your manuscript.

Response: This has been corrected.

Line 79-80: What is the full form on “(t-z)”? And what do you mean by N-Values. Please mention the full form and define these terms clearly for the readers to understand. For example, ‘N-Value’ should be ‘N-Value obtained from the Standard Penetration Test for the starta…’

Response: This has been corrected.

Line 165: The section “Site Overview” should be elaborated a bit more. In the current state the section does not really convey any description of the site, expect the visuals in Figure 3.

Response: This has been corrected.

Line 215: What is the full form on “BX”? When a short form is used for the first time in the manuscript, please mention the full form for the readers to understand.

Response: This has been corrected.

Table 7: The results presented in this table is the key information in this manuscript. However, it is fair to tabulate the result in a table and expect the reader to understand them completely. Authors SHOULD present a well structures discussion of the results and preferably compare with exiting research.

Response: This has been corrected.

Section 4: Authors do not explain the software program and procedure used in the numerical analysis. This should be clearly mentioned and discussed. The accuracy of the results depends on various steps used in numerical modelling, this is very important part. This SHOULD be revised.

Response: This has been corrected.

Reviewer 2 Report

This manuscript investigated the adhesion force applied with the concept of skin friction and pre-bored pile of drilled shaft according to domestic and foreign design standards This study is interesting with some novelties, which can be accepted after the following revisions:

  1. The clarity of some pictures needs to be improved such as Fig. 1 and 2.
  2. Material and size parameters of model need to be provided.
  3. How to monitor or identify the pull-out load? The recent advances on this filed can be referred: .
  4. Excessive conclusion results in the abstract, missing methods.
  5. What are the initial and boundary conditions in experiment?
  6. Which type of sensor selected in this work? How to place the sensors. The author can referred to the recent works on “A novel uncertainty-oriented regularization ……” and “A synchronous placement and size-based multi-objective optimization……”.
  7. Which parameter has the greatest impact on the result?

Author Response

Response to reviewer’s comment

Thank you so much for the reviewer’s invaluable comments on the paper. We revised the paper according to reviewer’s comments.

Reviewer 2

Comments and Suggestions for Authors

This manuscript investigated the adhesion force applied with the concept of skin friction and pre-bored pile of drilled shaft according to domestic and foreign design standards This study is interesting with some novelties, which can be accepted after the following revisions:

1. The clarity of some pictures needs to be improved such as Fig. 1 and 2.

Response : This has been corrected.

2. Material and size parameters of model need to be provided.

Response : This has been corrected.

3. How to monitor or identify the pull-out load? The recent advances on this filed can be referred: .

Response : It is mentioned in References 3.

4. Excessive conclusion results in the abstract, missing methods.

Response : This has been corrected.

5. What are the initial and boundary conditions in experiment?

Response: This has been corrected.

6. Which type of sensor selected in this work? How to place the sensors. The author can referred to the recent works on “A novel uncertainty-oriented regularization ……” and “A synchronous placement and size-based multi-objective optimization……”.

Response: In the lab experiment of this study, displacement was not measured using a sensor, but the pull-out force was checked using a load cell.

7. Which parameter has the greatest impact on the result?

Response: When designing piles, there is no design standard for estimating the pull-out force of pre-bored pile in rock, so the pull-out force calculation method of drilled shaft pile is generally used. The pull-out force of the pre-bored pile in rock is overestimated, and the problem of the pull-out of the pre-bored pile occurs frequently. Therefore, it is necessary to clearly confirm the skin friction embedded in the rock, which is related with the pull-out force of the pre-bored pile in rock.

Round 2

Reviewer 1 Report

The manuscript has been revised based on few of my comments provided in the initial review. However, there still are errors in the manuscript which needs careful attention. Below are my comment for author’s perusal.

Figure 1: There is a mistake in the redrawn figure. Refer to the zoomed section, the horizontal force should be “KP0” not “KPa”.

Figure 1 and Equation 1: the horizontal earth pressure coefficient is abbreviated as ‘K’ in the figure while as ‘Kp’ in the equation. This will confuse the readers. Authors are also advised to check through the entire manuscript again carefully for such errors.

Equation 3: the adhesive force in undrained condition is abbreviated as ‘c’ in the equation, while the description uses a new symbol of ‘cu’. This should be corrected. Authors are also advised to check through the entire manuscript again carefully for such errors.

Figure 6: Figure 6b is missing. Please check.

Author Response

Response to reviewer’s comment

Thank you so much for the reviewer’s invaluable comments on the paper. We revised the paper according to reviewer’s comments.

Reviewer 1

Comments and Suggestions for Authors

The manuscript has been revised based on few of my comments provided in the initial review. However, there still are errors in the manuscript which needs careful attention. Below are my comment for author’s perusal.

Figure 1: There is a mistake in the redrawn figure. Refer to the zoomed section, the horizontal force should be “KP0” not “KPa”.

Response: This has been corrected.

Figure 1 and Equation 1: the horizontal earth pressure coefficient is abbreviated as ‘K’ in the figure while as ‘Kp’ in the equation. This will confuse the readers. Authors are also advised to check through the entire manuscript again carefully for such errors.

Response: This has been corrected.

Equation 3: the adhesive force in undrained condition is abbreviated as ‘c’ in the equation, while the description uses a new symbol of ‘cu’. This should be corrected. Authors are also advised to check through the entire manuscript again carefully for such errors.

Response: This has been corrected.

Figure 6: Figure 6b is missing. Please check.

Response: This has been corrected.
